# Do Livelihood Strategies Influence Post-Epidemic Business Performance? Investigation of Social Capital and Environmental Perception

**Zhonghao Wang**

Applied Economics Department, Johns Hopkins University, Baltimore, MD 21218, USA; zwang246@163.com

**Abstract:** Livelihood strategies are a combination of activities and actions that individuals undertake to accomplish their desired goals. The current study aimed to examine the impacts of livelihood strategies on business performance. This study explored social capital as a mediator between livelihood strategies and business performance. This study further investigated how environmental perception strengthens livelihood strategies and business performance. For data collection, a quantitative research method and a random sampling technique were used. Data were collected through 550 questionnaires from small–medium enterprises (SMEs) in China. To test the study hypotheses, structural equation modeling (SEM) was performed. The findings confirm the notion that livelihood strategies are positively associated with business performance. The results also corroborate the notion that social capital mediates the association between livelihood strategies and business performance. The outcome validates the notion that livelihood strategies and business performance are further articulated when environmental perception is higher. Environmental perception moderated the impacts of livelihood strategies on business performance via social capital. The current research provides a better understanding of how livelihood strategies facilitate amplified business performance via social capital and the valuable role of environmental perception in research models.

**Keywords:** livelihood strategy; social capital; environmental perception; business performance; SMEs

## 1. Introduction

Based on a business performance framework, the current research examines the characteristics of livelihood strategies, social capital, and environmental perception in post-epidemic business performance. This paper expands on the findings of previous studies and evaluates the influences of all variables, i.e., the livelihood strategy, social capital, and environmental perception, on business performance. The word livelihood is frequently described as an individual's competence to earn and maintain a living wage. Moreover, livelihood strategies are a combination of activities or behaviors that people undertake to achieve livelihood goals and domestic survival [1]. However, livelihood strategies are divided into five core asset groups, i.e., human capital, physical capital, financial capital, social capital, and natural capital [2]. People seek out and combine diverse forms of capital as a livelihood strategy to survive [3]. In recent decades, the Food and Agriculture Organization of the UN (FAO), the United Nations Development Programme, and the Department for International Development (DFID) have made great efforts to analyze livelihood sustainability [4]. In addition, at present, the eye-catching link between livelihood strategies and business performance has attracted the attention of various scholars. Nowadays, SMEs have grown to become a significant sector that offers numerous livelihood strategies for individuals from developing nations and districts, thereby providing numerous alternative revenue sources for such individuals [5]. This also draws attention to the significant impacts of livelihood strategies on employees' social capital choices and business performance. SMEs, as providers of livelihood strategies, have gained the attention of researchers and scholars. A study on SMEs' livelihood strategies in various

nations and areas has highlighted that SMEs perform critical roles in decreasing poverty and improving individuals' livelihoods [6]. However, considering that each nation has distinctive development objectives, tourism resources, and economic levels, it follows that employees' domestic livelihood strategy choices and the numerous factors affecting the alternatives vary notably [7]. In addition to their economic effects, SMEs are also regarded as having positive influences on sustainable development. Regardless of the positive impacts of SMEs' livelihood strategies on their employees' poverty reduction, their positive impacts on business development and strategic performance have also been recognized, with their influence on business performance varying considerably among regions [8]. Social capital is a characteristic of livelihood strategies, and it comprises various factors, such as norms, social trust, and networks, which improve cooperation and management, thereby achieving shared benefits and facilitating better business performance [9]. The selection of a livelihood strategy helps in the sharing of knowledge, expertise, and information amongst individuals, and it is linked to their position in their social network, as well as contributing to the social capital that alternatively enhances the business performance of a firm [10]. Studies on business performance have also emphasized the significance of being proficient. A livelihood strategy comprises asset allocation and production activities and actions that people carry out to accomplish their business and family life goals [11]. Researchers have also shown that, in a firm, good business performance is mostly achieved by employees with the best livelihood strategies. Previous studies on the livelihood strategies of SMEs have focused primarily on internal mechanisms, for example, livelihood capital and geographical and strategy conditions [12]. However, an individual's psychological awareness dominates their performance and thoughts. Environmental perception is the understanding of how an individual exchanges and supplies information back and forth with their surrounding environment [13]. An individual's environmental perception has ethical and situational characteristics, and it reflects subjective, active responses and attitudes relating to natural environmental changes [14]. At present, environmental perception is a vital field of humanistic topography. This study explores the moderating role of environmental perception in the link between livelihood strategies and business performance. Environmental perception is the idea or awareness of worldwide ecological changes and their effects on an individual's livelihood [8]. A strong environmental perception gives an individual a vast responsibility for awareness of environmental changes, supporting their conscious ecological protection behavior and the utilization of natural green resources [15]. Previous studies have investigated business performance in various contexts, such as organizational learning [16], business intelligence [17], and e-marketing orientation [18]. To the best of our knowledge, the literature lacks a preceding study that mutually examines the effects of all these constructs on business performance. To bridge this study gap and guide management to understand the reasons behind successful business performance, this research model provides a unique perspective and aims to answer the following questions:

1. What is the impact of the livelihood strategy on business performance?
2. Does social capital mediate the link between alivelihood strategy and business performance?
3. Does environmental perception play a moderating role in the relationship between alivelihood strategy and business performance?

The remainder of this paper is organized as follows: Section 2 presents a literature review, and the following section, Section 3, details the methodology and related outcomes. Section 4 contains an analysis of the study variables, and the last section presents the discussion, implications, and future directions of this study, along with its limitations.

## 2. Literature Review

### 2.1. Livelihood Strategies and Performance

Livelihood strategies are a combination of actions that an individual carries out to accomplish their business goals. They include reproductive choices, creative activities, and investment strategies [19]. Livelihood strategies are a combination of resources that support,

control, and maintain different levels of business performance [20]. These benefits can be exchanged, applied, and gathered to create an income status that influences the economic behaviors of people and also enhance their business performance [21]. Efficient livelihood strategies assist in achieving desired business outcomes and creating a positive response loop that enhances the sustainable performance of business livelihood [22]. The fundamental idea of sustainable business performance is dependent upon five pillars, i.e., physical capital, human capital, financial capital, social capital, and natural capital [2]. With regard to post-epidemic business performance, the selection of a livelihood strategy is essentially established through business capital [2]. A livelihood strategy refers to a systematized set of lifestyle values, goals, activities, and choices that provide an income and the necessities of life. It encompasses the range of choices, actions, and activities made by a firm to achieve its desired goals [23]. The basic purpose of a good livelihood strategy is to decrease inequality and poverty by generating employment amongst poor individuals, resulting in highly susceptible households achieving sustainable living and businesses boosting their performance [24]. Firms develop livelihood strategies that personally enhance business performance. Business performance reflects a firm's ability to implement a strategy and utilize the resources required to accomplish organizational objectives [25]. A livelihood strategy comprises the way by which individuals combine their income-generating actions and behaviors; this includes the way by which assets are utilized, which resources are invested in, and how people are managed to protect accessible assets and earnings. These behaviors could help in enhancing and increasing sustainable business performance [26].

**H1:** *A livelihood strategy is associated with business performance.*

### 2.2. Mediating Role of Social Capital

Social capital is the societal position of an individual who is valuable to the growth of their family and who also assists in the development of resources [27]. A livelihood strategy aids an inventor in sharing information and knowledge concerning different projects; however, social capital is a key factor in the investment procedure, as the help of peers contributes to post-epidemic business performance [28]. Moreover, an individual's aptitude to select a livelihood strategy is closely linked to social capital and business performance. Improved access to livelihood strategies is conducive to developing adequate opportunities for employees to enhance their social capital and to improving business performance [2]. In rural regions, a social network mostly includes two components, i.e., a clan association network, shaped through blood links, and a resident's self-organization, structured with cooperative or trading relationships [29]. A livelihood strategy allows for the vendors of a project to add to their social capital via the development of strong ties with investors, suppliers, and community members, resulting in high business performance [30]. When business owners focus on the livelihood strategies of their employees and make strong social capital ties with them, business performance is likely to increase more effectively [31]. Improved livelihood strategies enable good business performance and increase investments inprojects. A social system widens a firm's access to information and experience, and it reduces reproduction costs [32]. Social capital refers to a system of relationships among diverse groups of people that allow people to collaborate in a group to efficiently achieve common goals. It is the potential capacity to acquire resources, information, and support, resulting in bonding and bridging among similar or diverse groups of people [33]. Business owners should focus on livelihood strategies and launch projects for employees and their social network to enhance business performance and investment funds for different projects [34]. Social capital is a bridge between livelihood strategies and business performance. Social capital supplies prospective resources that are accessible and obtained from within the network of relationships developed amongst individuals and communities. Depending on the livelihood strategy, a high level of social capital would increase the amount of information available and investment opportunities [35].

**H2:** *Social capital mediates between livelihood strategies and business performance.*

### 2.3. Moderating Role of Environmental Perception

Environmental perception is a critical factor that influences livelihood strategies and business performance. According to prior studies, environmental perception is the subjective attentiveness to surrounding environmental changes and ecological responsibility [36]. A livelihood strategy comprises the abilities, actions, and resources required to earn a living or achieve a high business performance [37]. A sustainable livelihood strategy can be developed in response to shocks and pressures, and it can maintain and strengthen individual capacities and skills without harming natural resource bases [38]. In a rational monetary community, individuals' behaviors are restricted by their capabilities, skills, knowledge, endowments, and other personal attributes, as well as by institutional policies and their external surroundings [39]. Moreover, researchers have found that an individual's actions are strongly linked to their psychological features, for instance, their environmental perceptions [40]. Environmental perception encompasses the awareness and feelings regarding the environment and the act of perceiving the environment through the senses. It is an idea applied to the association between individuals and societies within the same environment [41]. A livelihood strategy is an imperative dynamic force that can boost business performance [42]. Researchers have found that a livelihood strategy particularly influences an individual's motivation to participate in ecological protection policies that will increase firm performance [43]. Theoretically, an individual's significant choice of livelihood strategy may be affected by their environmental perception [42]. A livelihood strategy is based on the individual who maintains their fundamental economic interest, and ensuring conservation behavior will enhance business performance [44]. As mentioned above, environmental perception is an individual's awareness of environmental changes in their surroundings, and a high environmental perception will have a strong influence on livelihood strategies, as well as efficiently enhancing business performance; prior studies have found that individuals with a strong environmental perception have a stronger reliance on livelihood strategies that increase firm business performance [45]. Environmental perception is the psychological feature of an individual who willingly focuses on environmental protection measures, and moral standards that improve livelihood strategies and the self-concerns of an individual directly result in a high business performance [46].

**H3:** *Livelihood strategies and business performance are positively moderated by environmental perception.*

### 2.4. Framework

Figure 1 shows the theoretical framework.

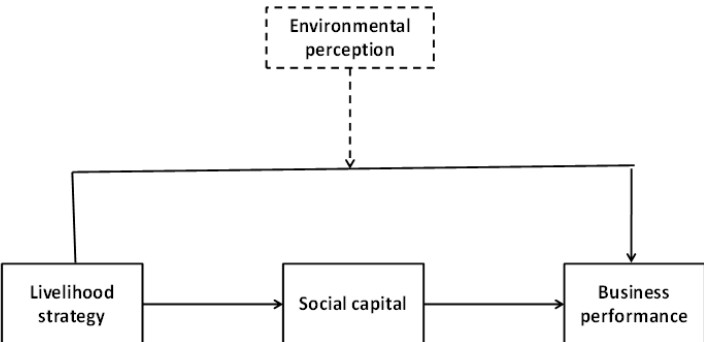

**Figure 1.** Theoretical framework.

## 3. Methodology

We examined the study hypotheses in targeted SMEs in China. For data collection, questionnaires were used as measures to assess the finding outcomes obtained through the random sampling method. It is suggested that these outcomes could be used for all SMEs and expanded on in the future.

### 3.1. Data Collection

For the collection of data, this study selected SMEs that are located in China. These firms are engaged in developing livelihood strategies, and they carry out seminars and training sessions to increase their social capital in order to enhance their post-epidemic business performance. They have also set up environmental perception programs in their firms to increase business performance. They offer services and design products that are eco-friendly and cost-effective. For data collection from managers, policy makers, and owners working in SMEs, we employed 550 questionnaires and distributed them in both hard and soft forms with the assistance of 5 research associates and the management firms. A translation procedure was employed to translate the questionnaires from English to Chinese and to re-translate them back into English with the assistance of three other experts. Afterward, all inconsistencies were settled by a discussion with experts; furthermore, a correspondence letter was attached to the questionnaires to explain the implications and the objectives of the current research. All participants were given the option to take part in the survey. Nevertheless, out of the total 550 questionnaires, we only received 392 that were completed and, thus, met the study criterion, and after further analysis, this was found to represent a 71.27% return rate. The remaining 152 questionnaires were disposed of due to in-completion. The questionnaire was divided into three sections: The first section contains the demographic details of the respondents, such as age, field experience, gender, and qualification of participants. The next section comprises the definitions of all study variable items. The last section includes the items of the study variables. Out of the total sample size, 65.83% of respondents were male, with an average age between 30 and 45 years; the remaining 34.17% of respondents were female, with an average age between 25 and 45 years. Furthermore, 53% of respondents had a bachelor's degree, 22.56% of respondents had a master's degree, and the remaining 24.44% of respondents had completed matric.

### 3.2. Measurement

A validity scale performs an extremely vital role in the design of survey tools. This study adapted pre-tested scales from prior empirical studies to ensure the validity and reliability of the present study. In the current research, 7-point Likert scales were used, with a range comprising 1 = strongly disagree, 3 = neutral, and 7 = strongly agree.

#### 3.2.1. Livelihood Strategy

Livelihood strategies were measured using a 5-item scale, which was adapted from [47]. This scale measures an individual's livelihood income stability, skills, expertise, and exposure to cultural and social norms and their link to well-being outcomes. An example item is "Our firm believes that it's significant for us to comply with SECP regulations".

#### 3.2.2. Social Capital

To measure social capital, a 4-item scale was used, which was adapted from [48]. This scale measures individual trust, social support, and associative relationships among employees in a firm. An example question is "In our firm we share same ambition, vision and values of helping each other to resolve problems regarding community".

#### 3.2.3. Business Performance

Business performance was measured using a 10-item scale, which was adapted from prior studies [18]. This scale measures metrics that determine whether a business has accomplished its desired goals in its planned time framework. A sample item is "Our firm accomplishes targets of the standard sales growth and effectiveness over last three years".

#### 3.2.4. Environmental Perception

For the measurement of environmental perception, a 6-item scale was used, which was adapted from [49]. This scale measures the environmental behavior and activities of

the firm. An example item is "Do you know what activities and things can be implemented to prevent environmental pollution?".

## 4. Results and Analysis

To assess the research hypotheses, we employed correlation, descriptive, and SEM (structural equation modeling) techniques. This research employed "Process" software, developed by [50], to analyze the mediation function of social capital. With the use of their method, [51] assessed discriminate validity. Using Cronbach's alpha values, the model's validity was examined. The research model fits the data, according to the CFA findings.

To examine the model's suitability and the scales' validity, CFA was employed. Each of the validity scales discriminates, converges, and predicts–performs as expected (see Table 1). The results show that the model's reliability ranged from 0.70 to 0.91, proving that the model used in the present research is reliable. Through CFA, the discriminant validity and CR were shown, and factor loading was higher than 0.70. The [51] method was also employed in this research to analyze the AVE, and the results show that AVE > 0.50, while alpha > 0.70.

**Table 1.** Validity results.

|  | Items | Alpha | F. Loading | CR | AVE |
|---|---|---|---|---|---|
| Livelihood strategy | 5 | 0.82 | 0.73–0.91 | 0.87 | 0.68 |
| Social capital | 4 | 0.79 | 0.70–0.88 | 0.92 | 0.71 |
| Environmental perception | 6 | 0.86 | 0.76–0.90 | 0.94 | 0.73 |
| Business performance | 10 | 0.81 | 0.71–0.93 | 0.90 | 0.69 |

Table 2 presents the results of the model fit, and the four-factor model confirms the satisfactory results (RMSEA = 0.05, $\chi^2$ = 1011.14, df = 390; $\chi^2$/df = 2.592; CFI = 0.92; GFI = 0.91).

**Table 2.** CFA results.

| Model Details | $\chi^2$ | Df | $\chi^2$/df | RMESA | GFI | CFI |
|---|---|---|---|---|---|---|
| Hypothesized four-factor model | 1011.14 | 390 | 2.592667 | 0.05 | 0.91 | 0.92 |
| Three-factor model | 1243.51 | 350 | 3.552886 | 0.15 | 0.88 | 0.89 |
| Two-factor model | 1348.47 | 360 | 3.74575 | 0.22 | 0.71 | 0.72 |
| Single-factor model | 1298.22 | 290 | 4.476621 | 0.26 | 0.64 | 0.65 |

### 4.1. Correlations

Table 3 shows the correlations, and they prove our theory and demonstrate associations among all the variables. The livelihood strategy is positively associated with social capital (r = 0.32 **, $p$< 0.0001) and business performance (r = 0.21 **, $p$ = sig). Social capital is positively associated with business performance (r = 0.30 **, $p$ = sig) and environmental perception (r = 0.23 **, $p$ = sig). Similarly, environmental perception is positively associated with business performance (r = 0.17 **, $p$ = sig).

### 4.2. Hypothesis Testing

The SEM technique was used to test the study hypotheses (see Table 4). The livelihood strategy predicts business performance (Beta = 0.22 **, $p$ = sig); thus, H1 is confirmed. The livelihood strategy is significantly linked to social capital (Beta = 0.33 **, $p$= sig); H2 is confirmed. Social capital is significantly related to business performance (Beta = 0.32 **, $p$ = sig); therefore, H3 is confirmed.

**Table 3.** Correlations.

| Constructs | Mean | SD | 1 | 2 | 3 | 4 | 5 | 6 | 7 | 8 |
|---|---|---|---|---|---|---|---|---|---|---|
| Res. Gender | 0.9 | 0.81 | 1 | | | | | | | |
| Res. Age | 33 | — | 0.09 | 1 | | | | | | |
| Experience | 2.9 | 0.84 | 0.08 | 0.03 | 1 | | | | | |
| Education level | 2.4 | 0.91 | 0.06 | 0.05 | 0.04 | 1 | | | | |
| Livelihood strategy | 3.8 | 0.93 | 0.09 | 0.12 * | 0.08 | 0.07 | 1 | | | |
| Social capital | 3.5 | 0.91 | 0.05 | 0.09 | 0.04 | 0.05 | 0.32 ** | 1 | | |
| Business performance | 3.9 | 0.95 | 0.03 | 0.07 | 0.06 | 0.09 | 0.21 * | 0.30 ** | 1 | |
| Environmental perception | 3.6 | 0.90 | 0.08 | 0.03 | 0.04 | 0.09 | 0.25 ** | 0.23 * | 0.17 * | 1 |

Note: * $p < 0.5$, ** $p < 0.1$. standard deviation (SD).

**Table 4.** Results of hypotheses.

| Details | Effects | Coefficient | Remarks |
|---|---|---|---|
| (H1) Livelihood strategy → business performance | + | 0.22 ** | Accepted |
| (H2) Livelihood strategy → social capital | + | 0.33 ** | Accepted |
| (H3) Social capital → business performance | + | 0.32 ** | Accepted |

** $p < 0.1$.

### 4.3. Mediating Role of Social Capital between Livelihood Strategy and Business Performance

Table 5 presents the mediation of social capital between the livelihood strategy and business performance. To conduct the mediating test, we followed the instructions of Preacher and Hayes's approach [50]. Using their approach, the social capital mediation was validated, with a considerable indirect effect. The findings illustrate that social capital performs mediation between the livelihood strategy and business performance (B = 0.1321, low = 0.1783, Up = 0.1325). We also utilized the Sobel test/"Z score", and the outcomes confirm that the Z score = 5.53 ** was significant.

**Table 5.** Results of indirect effects of livelihood strategy.

| Model Detail | Data | Boot | Bias | SE | Lower | Upper |
|---|---|---|---|---|---|---|
| LS → SC → BP | 0.1321 | 0.1572 | −0.0005 | 0.259 | 0.1783 | 0.1325 |
| Soble Test Z Score = 5.53 ** | | | | | | |

Note: LS, livelihood strategy; SC, social capital; BP, business performance ** $p < 0.1$.

### 4.4. Moderating Role of Environmental Perception on LS and BP Link

A hierarchical regression analysis was conducted in order to observe the moderation effect of environmental perception on the association between SC and BP. The findings of the hierarchical regression analysis are presented in Table 6 using Step 1, Step 2, and Step 3 procedures. The base model information is depicted in the Step 1 and Step 2 columns in Table 6. Furthermore, the Step 3 column presents the coefficients of the moderating effect of environmental perception on the relationship between SC and BP. Table 6 also presents the coefficient of the interaction term, i.e., SCx environmental perception, which indicates that environmental perception positively affects the connection between SC and BP (β = 0.24, $p < 0.01$). As per the suggestion of [52], we also conducted a slope analysis.

**Table 6.** Results of moderation.

| | Step 1 | Step 2 | Step 3 |
|---|---|---|---|
| Moderation of frugal innovation | | | |
| Res. gender | 0.028 | 0.010 | 0.009 |
| Res. age | 0.023 | 0.020 | 0.017 |
| Experience | 0.007 | 0.005 | 0.006 |
| Educational level | 0.033 | 0.034 | 0.043 |

**Table 6.** *Cont.*

|  | Step 1 | Step 2 | Step 3 |
|---|---|---|---|
| Social capital |  | 0.30 ** | 0.33 ** |
| Environmental perception |  | 0.22 ** | 0.26 ** |
| Social capital x environmental perception |  |  | 0.24 ** |
| $R^2$ | 0.009 | 0.191 | 0.198 |
| Adjusted $R^2$ | 0.003 | 0.159 | 0.175 |
| $\Delta R^2$ | 0.007 | 0.163 | 0.028 |
| $\Delta F$ | 4.172 | 79.63 | 17.13 |

** $p < 0.1$.

## 5. Discussion

This study investigates how a firm's livelihood strategy influences its business performance through sense-making practices drawing on the social individuality theory. In this research, H1 is confirmed, demonstrating that the livelihood strategy is directly associated with business performance. The H1 outcomes are congruent with prior findings demonstrating that livelihood strategies are a combination of the actions that an individual carries out to accomplish business goals. Livelihood strategies include reproductive choices, creative activities, and investment strategies [19]. Livelihood strategies are a combination of resources that support, control, and maintain different levels of business performance [20]. These benefits can be exchanged, applied, and gathered to create an income status that influences the economic behaviors of people and also enhance their business performance [21]. Efficient livelihood strategies assist in achieving desired business outcomes, and they create a positive response loop that enhances business livelihood sustainability performance [22]. The outcomes confirm H1, demonstrating that a livelihood strategy is directly associated with business performance. In addition to considering direct outcomes, the current study's aim was to determine the underlying factors that are responsible for a firm's livelihood strategy positively affecting its business performance. Thus, in H2, we proposed that the link between livelihood strategies and business performance was mediated by social capital. This research explores the indirect impact of social capital on the relationship between livelihood strategies and business performance, and optimism was found to increase the influence of livelihood strategies on business performance via social capital. A firm's social capital is determined through its environmental perception, and this is linked to its livelihood strategies, social obligations, and corporate behaviors, instead of company expertise, in product and service advancement. The findings regarding H2 support those of previous studies, wherein social capital was found to encompass the societal position of an individual who is valuable to the growth of their family and who assists in the development of resources [27]. A livelihood strategy aids its creator in sharing information and knowledge concerning different projects; however, social capital is a key factor in the investment procedure that adds to post-epidemic business performance with the help of peers [28]. Moreover, an individual's aptitude to select a livelihood strategy is closely linked to social capital and business performance. Improved access to livelihood strategies is conducive to developing adequate opportunities for employees to enhance their social capital and improve business performance [2]. Specifically, the outcomes of this study confirm H2, demonstrating that social capital plays a dominant mediating role in the relationship between livelihood strategies and business performance. Additionally, environmental perception adds to the positive link between livelihood strategies and business performance. Furthermore, environmental perception further increases the indirect influence of livelihood strategies and business performance via social capital. This study's outcomes add to the increasing body of literature, striving to elucidate the psychological mechanism underlying environmental perception. The current research emphasizes the significance of a firm's environmental perception with respect to CSR activities and livelihood strategies for enhancing business performance. In this study, H3 states that environmental perception moderates the relationship between livelihood strategies and business performance.

The results are consistent with those of previous studies. Environmental perception is a critical factor that influences livelihood strategies and business performance. According to previous studies, environmental perception is the subjective attentiveness to surrounding environmental changes and ecological responsibility [36]. Livelihood strategies comprise the abilities, actions, and resources required to earn a living or achieve a high business performance [37]. A sustainable livelihood strategy can develop in response to shocks and pressures, and it can maintain and strengthen individual capacities and skills without harming natural resource bases [38]. The findings corroborate the notion that environmental perception plays a moderating role in the link between livelihood strategies and business performance. Overall, this empirical model contributes to the understanding of how, when, and why livelihood strategies, social capital, and environmental perception affect the business performance of a firm.

## 5.1. Theoretical Implications

The current study contributes to the literature in various ways. Firstly, this study extends the research on both livelihood strategies and business performance, as it aimed to relate livelihood strategies to business performance via sense-making practices. However, previous studies consider livelihood strategies and business performance from different perspectives. This study proposes that a firm's livelihood strategy affects its employees' attitudes and actions, thus increasing business performance. This study's findings also support those of previous studies that emphasized different conceptualizations of livelihood strategies. Secondly, the current research broadens previous studies' findings by identifying the mechanisms underlying how livelihood strategies directly enhance business performance by integrating social capital as a mediator. This study's findings are in agreement with those of previous studies, where livelihood capital was found to influence livelihood strategies by generating social capital. The outcomes of our research support the social identity theory, as they elucidate the mediation role of social capital in the link between livelihood strategies and business performance. Lastly, environmental perception was considered a factor of corporate awareness and responsibility in previous studies. However, the current research shows that environmental perception plays a moderating role in the link between livelihood strategies and business performance. Employees acting in accordance with the livelihood strategies of a firm are more likely to improve business performance when they have a high environmental perception. Additionally, environmental perception strengthens the indirect influence of livelihood strategies on business performance, and it mediates via social capital. Our study outcomes also emphasize the role of environmental perception as a significant moderator in elucidating the relationship between livelihood strategies, social capital, and improved business performance.

## 5.2. Practical Implications

This study has valuable implications for management, policy makers, and practitioners. Firstly, this study's outcomes show that livelihood strategies affect business performance; thus, SMEs should consider their livelihood strategies to be valuable promotional strategies that not only improve their brand or performance but also generate social capital. Business livelihood capital could act as a tool to support livelihood strategies. Hence, management should develop and communicate livelihood strategies that improve business performance. Secondly, employees feel attached to a firm that fulfills the values desired by stakeholders. Thus, firms should use their resources on livelihood strategy initiatives, for instance, environmental protection, redevelopments, greening campaigns, and programs raising awareness of ecological changes, to increase social capital and stimulate employees' willingness to participate in activities that enhance business performance. Social capital can result in the development of strong relationships that provide long-term competitive benefits, as social capital mediates the link between livelihood strategies and business performance. Thirdly, scholars should acknowledge that the positive impacts of livelihood strategies on business performance are greatly influenced by the environmental perceptions

of the firm. Environmental perception changed positive associations between livelihood strategies and business performance. Thus, management must pay further attention to environmental perception actions in a credible way to facilitate the development of good livelihood strategies that alternatively increase business performance.

*5.3. Limitations and Future Research*

This study's contribution must be considered in light of the following drawbacks, some of which provide directions for future studies: Firstly, the quantitative method and the random sampling data interfered with the strong fundamental assumptions regarding the mediation link. We recommend future studies carry out longitudinal and cross-sectional designs to gain a better grasp of social capital. Secondly, we note that this study only collected data from SMEs in China, which restricts its external validity. In future studies, this empirical model should be examined in other contexts, such as different industries, tourism sectors, and cultures; examining this empirical model in other regions and nations would be preferable to confirm the external validity of the findings. Thirdly, we employed a single mediator, i.e., social capital, to describe the association between livelihood strategies and business performance. Upcoming studies should consider organizational readiness as a mediator and AI adoption as a moderator in the enhancement of business performance through the antecedents of livelihood strategies. Finally, the current study's results might need to be repeated with firm customers. As customers are external to a firm, to test the external validity, we should determine whether these findings can be replicated. Moreover, future studies must also investigate the differences between employees and a firm's customers.

**Funding:** This research received no external funding.

**Institutional Review Board Statement:** The study was conducted in accordance with the declaration of the Hopkins Applied Research Board, Baltiomor (Project No.20BTJ031).

**Informed Consent Statement:** Written informed consent has been obtained from the patient(s) to publish this paper.

**Data Availability Statement:** The raw data that support the conclusions of the current article will be made available upon request.

**Conflicts of Interest:** The author declares no conflict of interest.

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
