# Peer review of "Do Livelihood Strategies Influence Post-Epidemic Business Performance? Investigation of Social Capital and Environmental Perception"

_sustainability, doi:10.3390/su15054532_

Round 1

Reviewer 1 Report

In the abstract we would prefer avoid double usage of "also" and have instead: also explore(s) .................. and investigate(s) .................

It is necessary to explain in abstract the SME means small and medium enterprises. To do it, we firstly write the collocation in full, then the acronym.

For the information of the author: it is not acceptable if you are referring to a source without mentioning the  scholar, the data, the book but only giving a digit, like [1], [2], [3].  No reader should suffer and turn pages up and down. Instead, the text should have said, e.g.,  Liu, W., et al. [1] mention/state/claim/urge that livelihood strategies mostly refer to behaviors and actions undertaken for the domestic survivals.

This remark goes to the entire text.

line 29-32 we would like to understand who paid attention: either they were Food-Agriculture institute of UN (FAO), the United-Nations Development programs and Department for International-Development (DFID). Then they have paid attention to sth, or these organizations HAVE been paid attention for analysis. Also, put a comma before FOR INSTANCE. 

line 33 passive/active voices are confused (as on line 29-32) and the grammar tense

line 36 grammar

We would suggest the author(s) reconsider the text as it is not readable. Sometimes it is not even easy to follow the logic: are all these scholars and sources that you have mentioned, supporting the same idea, or they are arguing with each other? The work is just an encyclopaedia of thoughts but it does not give the full understanding.

Another point is grammar/tense errors, punctuation, article errors making the further reading impossible, e.g., SMEs performs (if SMEs are plural here, the only thing they can and must do is SMEs PERFORM).

Author Response

Reviewer-1

Suggestion Incorporated (Green Highlighted part)

In the abstract we would prefer avoid double usage of "also" and have instead: also explore(s) .................. and investigate(s) .................

It is necessary to explain in abstract the SME means small and medium enterprises. To do it, we firstly write the collocation in full, then the acronym.

For the information of the author: it is not acceptable if you are referring to a source without mentioning the  scholar, the data, the book but only giving a digit, like [1], [2], [3].  No reader should suffer and turn pages up and down. Instead, the text should have said, e.g.,  Liu, W., et al. [1] mention/state/claim/urge that livelihood strategies mostly refer to behaviors and actions undertaken for the domestic survivals.

This remark goes to the entire text.

line 29-32 we would like to understand who paid attention: either they were Food-Agriculture institute of UN (FAO), the United-Nations Development programs and Department for International-Development (DFID). Then they have paid attention to sth, or these organizations HAVE been paid attention for analysis. Also, put a comma before FOR INSTANCE. 

line 33 passive/active voices are confused (as on line 29-32) and the grammar tense

line 36 grammar

We would suggest the author(s) reconsider the text as it is not readable. Sometimes it is not even easy to follow the logic: are all these scholars and sources that you have mentioned, supporting the same idea, or they are arguing with each other? The work is just an encyclopaedia of thoughts but it does not give the full understanding.

Another point is grammar/tense errors, punctuation, article errors making the further reading impossible, e.g., SMEs performs (if SMEs are plural here, the only thing they can and must do is SMEs PERFORM).

       Your suggestion is incorporated.

SMEs abbreviation is explained in abstract.

Your suggestion is incorporated.

This paragraph is rephrased and improved.

Your all suggestions are incorporated.

Many sentences are rephrased. You can check the highlighted paragraph in introduction section.

Your suggestion is integrated.

Reviewer 2 Report

This is an interesting manuscript that aims to explore the impact of livelihood strategy on business performance, as well as the mediating role of social capital and the moderating role of environmental perception in the Chinese context. While I think that the research question is relevant and interesting, I also believe that the following points must be addressed:

1) The introduction is very long. I suggest moving some of the literature related to livelihood strategy, business performance, and social capital to the literature review section.

2) In the literature review, the author(s) must further explain the concepts of livelihood strategy (2.1), social capital (2.2), environmental perception (2.3), and business performance (2.4). I recommend that, in order to develop the literature review, the author(s) integrate(s) each concept into the statement supporting the corresponding hypothesis. For example, the descriptions of livelihood (2.1) and business performance (2.4) should be merged with the section of livelihood strategy and performance (2.5).

3) The author(s) should strengthen the discussion of the study’s findings. The current discussion of the implications appears somewhat cursory. It is important to consider what can be done with the study’s empirical evidence. Finally, the author(s) should provide more specific managerial implications.

Author Response

Reviewer-2

Suggestion Incorporated (Please refer to Blue Highlighted Part)

Comments and Suggestions for Authors

This is an interesting manuscript that aims to explore the impact of livelihood strategy on business performance, as well as the mediating role of social capital and the moderating role of environmental perception in the Chinese context. While I think that the research question is relevant and interesting, I also believe that the following points must be addressed:

1) The introduction is very long. I suggest moving some of the literature related to livelihood strategy, business performance, and social capital to the literature review section.

2) In the literature review, the author(s) must further explain the concepts of livelihood strategy (2.1), social capital (2.2), environmental perception (2.3), and business performance (2.4). I recommend that, in order to develop the literature review, the author(s) integrate(s) each concept into the statement supporting the corresponding hypothesis. For example, the descriptions of livelihood (2.1) and business performance (2.4) should be merged with the section of livelihood strategy and performance (2.5).

3) The author(s) should strengthen the discussion of the study’s findings. The current discussion of the implications appears somewhat cursory. It is important to consider what can be done with the study’s empirical evidence. Finally, the author(s) should provide more specific managerial implications.

Thank you for supporting our work. It is nice to read some positive comments from the reviewer.

Your suggestion is incorporated.

Thanks for guiding us; your suggestions are incorporated. You can check the highlighted section.

Discussion is improved.

Round 2

Reviewer 1 Report

Dear authors

It is now a new work: clear and better. It is readable and logical.
